# TOPLOC: A Locality Sensitive Hashing Scheme for Trustless Verifiable Inference

**Jack Min Ong**[1]   **Matthew Di Ferrante**[1]   **Aaron Pazdera**[1]   **Ryan Garner**[1]   **Sami Jaghouar**[1]   **Manveer Basra**[1]
**Max Ryabinin**[2]   **Johannes Hagemann**[1]

## Abstract

Large language models (LLMs) have proven to be very capable, but access to frontier models currently relies on inference providers. This introduces trust challenges: how can we be sure that the provider is using the model configuration they claim? We propose TOPLOC, a novel method for verifiable inference that addresses this problem. TOPLOC leverages a compact locality-sensitive hashing mechanism for intermediate activations, which can detect unauthorized modifications to models, prompts, or precision with 100% accuracy, achieving no false positives or negatives in our empirical evaluations. Our approach is robust across diverse hardware configurations, GPU types, and algebraic reorderings, which allows for validation speeds significantly faster than the original inference. By introducing a polynomial encoding scheme, TOPLOC minimizes the memory overhead of the generated proofs by $1000\times$, requiring only 258 bytes of storage per 32 new tokens, compared to the 262 KB requirement of storing the token embeddings directly for Llama 3.1-8B-Instruct. Our method empowers users to verify LLM inference computations efficiently, fostering greater trust and transparency in open ecosystems and laying a foundation for decentralized, verifiable and trustless AI services.

## 1. Introduction

In recent years, large language models (LLMs) have transformed natural language processing, enabling capabilities such as high-quality text generation, advanced dialogue systems, and improved reasoning (Grattafiori et al., 2024; Gemma Team et al., 2024). Inference providers, entities that run open-weights LLMs on their own hardware and expose model outputs via APIs, have risen to meet the demands of users who lack the resources or expertise to operate large-scale inference pipelines themselves.

However, a critical challenge arises in this open ecosystem: trust. Users must trust that an inference provider is faithfully serving the model as advertised, without undisclosed modifications. A provider could, for instance, secretly reduce numerical precision to cut costs, fine-tune the model to introduce certain biases, or prepend an undisclosed system prompt to steer the model's outputs. Without robust verification methods, users can only rely on the provider's claims, leaving them vulnerable to having their outputs tampered with (Chen et al., 2023). There is thus a need for **verifiable inference** — methods of verifying that a certain model and prompt were used during the inference computation.

A standard approach to perform this verification is to use cryptographically verifiable computing methods (Sun et al., 2024a; Modulus Labs, 2023; Kang et al., 2022; Sun et al., 2023; Ghodsi et al., 2017). However, they are either restrictive in the operations that are supported such that LLMs cannot be used or are currently too computationally expensive to be practically applied to LLM inference.

Another approach is to record the model's intermediate activation tensors during inference (Sun et al., 2024b; Zhang et al., 2024). By making these activations available, a third party could independently rerun the model on the same inputs and verify that the intermediate computations match, thus ensuring the authenticity of inference. However, direct storage of these intermediate tensors as the proof can be prohibitively expensive at scale. For example, storing the last hidden activations of 100,000 queries consisting of 4096 tokens for Llama 3.1-70B would take 6.7 terabytes.

In this work, we propose TOPLOC, an inference verification method that can reduce the storage cost of the proof by more than **1000x** while still maintaining the security guarantees of checking intermediate activations. The method performs locality-sensitive hashing of the intermediate activations, which encodes the top-k values and indices as a polynomial congruence. This polynomial congruence can then be compared against a tensor obtained from recomputation. Our method is also robust to nondeterminism of GPU operations and algebraic reorderings of the computation, which allows

[1]Prime Intellect [2]Together AI. Correspondence to: Johannes Hagemann <johannes@primeintellect.ai>.

the validation of the proof to be done significantly faster than the original inference.

Our contributions are as follows:

- We present a novel model inference hashing method called TOPLOC, which is easy to implement in modern inference engines with minimal overhead.

- We show that it is capable of detecting when a different model, prompt, or precision was used with 100% accuracy in our experiments.

- We verify that the method is robust to reordering of the computation caused by different GPU types, tensor parallel dimensions, as well as different attention kernel implementations.

- We propose a method of reducing the memory requirements of storing comparable points by encoding them as a polynomial, requiring a proof size of only 258 bytes for every 32 new tokens generated.

## 2. Related Work

Numerous methods have been proposed to verify the correctness of LLM inference performed by untrusted entities. The methods can be categorized into cryptographic verifiable computing and activation-based validation.

**Cryptographic Verifiable Computing.** Cryptographic Verifiable Computing allows one to verify that a computation was performed correctly on an untrusted computing provider using mathematical proofs. These techniques have been applied to machine learning models and neural networks (Sun et al., 2024a; Modulus Labs, 2023; Sun et al., 2023; Kang et al., 2022; Ghodsi et al., 2017). However, most of them require the computations to be expressed as an arithmetic circuit, limiting the functions that can be used in the model. The translation of the model to an arithmetic circuit also hurts the model quality and makes the proof generation schemes unable to utilize optimized inference engines such as vLLM[1], TensorRT[2], and SGLang[3].

Moreover, the size of modern LLMs introduces substantial computational overhead for for these methods. zkLLM (Sun et al., 2024a) takes 986 seconds to generate a proof that takes 803 seconds to validate for each inference computation for LLaMa 2-13B. This would mean that a single request that returns 2000 new outputs tokens would require about 23 days to generate the proof for and then 18 days to validate.

**Activation-based validation.** SVIP (Sun et al., 2024b) proposes training a proxy model to map the relationship be-

tween the final layer activations and labels derived from the inference input to produce a fingerprint and then mitigate the ability of an attacker to reverse engineer the proxy model by regularly rotating a secret in the form of an appended vector. However, the security of the scheme requires retraining of the proxy model by a trusted third party. Moreover, it also requires that the providers cannot obtain the client secret, which they may obtain by also being a client themselves.

Verisplit (Zhang et al., 2024) proposes the construction of a Merkle tree hash compression of a portion of the activations. However, this compression utilizes a cryptographic hash function, rendering it incompatible with nondeterministic computation caused by GPU kernel scheduling, as well as algebraic reordering of the computations.

## 3. Background

### 3.1. Inference Modifications

Inference providers often make adjustments to computation methods to optimize for cost, efficiency, or specific commercial goals. While these modifications can make inference more economical and scalable, they may also impact the quality, and transparency of the service provided to users.

**Lower precision.** Inference providers might use lower precision formats, such as fp8 (Micikevicius et al., 2022) or bf16 (Kalamkar et al., 2019), which significantly reduces the inference compute and memory requirements.

**KV cache compression.** Intermediate tensors can be compressed to enable longer and faster generations with a slightly reduced response quality (Shi et al., 2024).

**Altered model weights.** Providers may distill, merge, or prune weights to reduce compute and memory requirements.

**Altered system prompt.** Providers could modify the system prompt to align with their commercial goals, incorporate specific biases, or prioritize certain outcomes.

### 3.2. Nondeterminism in Model Inference on GPU

Nondeterminism in computations performed on GPUs can arise from operation scheduling and differences in how intermediate results are handled (Monniaux, 2008; Whitehead & Fit-Florea, 2011).

Discrepancies in GPU computations can also arise from algebraic rewrites of the computation. These rewrites are often employed to improve the computational intensity of scheduled kernels, increase efficiency, and allow for parallelization across multiple GPUs.

Furthermore, there are several other causes for variability in computation results when running large language models on GPUs. For example, different GPU models often differ in

---

[1]`github.com/vllm-project/vllm`
[2]`github.com/NVIDIA/TensorRT`
[3]`github.com/sgl-project/sglang`

hardware architecture and precision handling. In addition to that, the CUDA versions determine the libraries and kernels used during computation, and differences in those versions can affect inference. Moreover, subtle variations in how the attention mechanism and other layers are implemented can introduce subtle numerical discrepancies. Lastly, the partitioning and aggregation of tensors when using tensor parallelism can also cause numerical deviations.

While these numerical discrepancies may seem negligible, they can have a large cascading effect across the long sequence of computations in model inference. This amplification can cause subtle but meaningful variations in the model's output, making reproducibility and consistency of results a significant challenge.

### 3.3. Source of Error in Transformer Models

In bf16 computations of transformers (Vaswani et al., 2017), most errors arise from the appearance of exact zeros. These zeros are the result of catastrophic cancellations (Goldberg, 1991) within the residual stream of the attention layer. Because the matrix multiplications involved in attention and MLP layers are nondeterministic (Golden et al., 2024), the occurrence of these exact zeros can be nondeterministic.

Interestingly, this behavior reveals a notable property: small values are more susceptible to rounding errors, whereas larger values tend to be represented consistently after algebraic reordering. This insight motivates us to focus on the larger values in a tensor when designing the hash function. By prioritizing large values, we can reduce the impact of rounding errors and improve the robustness of the hash function. We verify this empirically in Section 5.3.

## 4. The TOPLOC Algorithm

The TOPLOC algorithm encodes and validates the most salient features of a hidden state tensor using a compact, verifiable proof, as detailed in Algorithms 1 and 2.

Storing top-$k$ indices and values directly is inefficient, requiring 6 bytes per point: 4 for the index and 2 for the value. However, we can maintain comparability while storing less by interpolating a polynomial that passes through the points generated by the top-k indices and values. Given $k$ points, there always exists a unique $k-1$-degree polynomial that goes through the points which can be represented by $k$ coefficients. We thus only need to store the $k$ coefficients for the proof, each of which consists of 2 bytes.

In order to avoid floating-point issues with the polynomial, we interpolate a polynomial congruence in the integer field instead of in the real numbers. However, this means we need to find a reproducible unique mapping of the x values into the modulus group, as we cannot interpolate a polynomial that yields different $y$ values for the same value of $x$.

We thus need to find a modulus, $m$, such that the function $f(x) = x \mod m$ is injective on the set of indices.

During proof generation (Algorithm 1), the top-$k$ indices and values are calculated and an injective modulus $m$ is computed to uniquely map the indices. The indices and their corresponding values are encoded into a polynomial, which, along with the modulus, forms the proof.

---

**Algorithm 1** TOPLOC Proof Generation Algorithm

1: **Input:** Hidden state tensor $h$, top-$k$ parameter $k$
2: **Output:** Encoded proof $p$
3:
4: $(i, v) \leftarrow \texttt{topk}(h, k)$ {Find top-$k$ indices and values}
5: $m \leftarrow \texttt{findInjectiveModulus}(i)$
6: $i_m \leftarrow i \mod m$
7: $P(x) \leftarrow \texttt{InterpolateModPolynomial}(i_m, v)$
8: $p \leftarrow \texttt{encode}(m, P(x))$

---

For validation (Algorithm 2), the proof is decoded to retrieve $k$, $m$, and the polynomial. The top-$k$ features are recalculated and compared against the proof by checking for differences in the exponent and mantissa. The validation succeeds if the number of exponent mismatches, mean mantissa differences and median mantissa differences are below a set of thresholds.

---

**Algorithm 2** TOPLOC Proof Validation Algorithm

1: **Input:** Hidden state tensor $h$, encoded proof $p$
2: **Output:** Boolean validity flag $v$
3:
4: $k, m, P(x) \leftarrow \texttt{decode}(p)$
5: $(i, v) \leftarrow \texttt{topk}(h, k)$ {Find top-$k$ indices and values}
6: $i_m \leftarrow i \mod m$
7: $(err_e, err_m) \leftarrow (0, [])$
8: **for** $j = 1$ to $k$ **do**
9: $\quad (e_p, m_p) \leftarrow \texttt{extractBits}(P[i_m[j]])$
10: $\quad (e_v, m_v) \leftarrow \texttt{extractBits}(v[j])$
11: $\quad$ **if** $e_p = e_v$ **then**
12: $\quad\quad err_m.\texttt{append}(\|m_p - m_v\|)$
13: $\quad$ **else**
14: $\quad\quad err_e \leftarrow err_e + 1$
15: $\quad$ **end if**
16: **end for**
17: **if** $err_e > T_{\exp}$ **or** $\texttt{mean}(err_m) > T_{\text{mean}}$ **or** $\texttt{median}(err_m) > T_{\text{median}}$ **then**
18: $\quad$ **return** $False$
19: **end if**
20: **return** $True$

---

Appendix C details the subroutines and their respective theoretical computational complexities. The corresponding implementations are also available on GitHub[4].

---

[4] github.com/PrimeIntellect-ai/toploc

*Figure 1.* When generating the response, we need to perform one prefill for the input tokens and then multiple decodes for each new token generated. When validating, we can pass all the tokens at once and perform just one prefill. Decodes are not efficient on GPUs because they are memory-bound (Agrawal et al., 2024). Notice that the sequential nature of generation causes us to need the decoder blocks multiple times in the generation computation. This increases the total amount of data movement required to perform the computation. The decodes are thus bottlenecked by the time it takes to move data from GPU HBM (High Bandwidth Memory) to SRAM (Shared Memory).

## 5. Experimental Validation

### 5.1. Dataset and Models

For our experiments, we use the UltraChat dataset (Ding et al., 2023). The UltraChat dataset contains 1.4 million dialogues consisting of real-world inquiries, creative writing prompts, and various other text-based tasks such as rewriting, continuation, summarization, and inference, covering a wide range of topics.

We conduct experiments with three models: Llama 3.1-8B-instruct (Grattafiori et al., 2024), INTELLECT-1-instruct (Jaghouar et al., 2024), and Gemma-2-9b-it (Gemma Team et al., 2024), aiming to capture diversity across different architectural dimensions. Llama-3.1-8B-instruct and Intellect-1-instruct share a similar transformer block architecture but differ in the number of layers, while Gemma-2-9b-it features a different hidden dimension and MLP activation function.

### 5.2. Experiment Setup

We use the bf16 precision for all our experiments, unless specified otherwise. bf16 is commonly used in practice for activations in language model inference. However, compared to fp16 and fp32 precision, it is most prone to catas-

trophic cancellations. It contains only 7 bits of mantissa, as opposed to 23 mantissa bits for fp32 and 10 bits for fp16.

For all of our experiments, we perform the generation autoregressively, generating each new token separately with KV caching. During validation, we obtain the last hidden state activations for all tokens at once. This allows the validation to be done significantly faster than the generation, as the prefill is significantly more compute-intensive than the autoregressive decoding and is thus able to better utilize the GPU resources (Agrawal et al., 2024; 2023).

For thresholds, we use $T_{exp} = 38$, $T_{mean} = 10$ and $T_{median} = 8$ for bf16 inference and $T_{exp} = 8$, $T_{mean} = 256$ and $T_{median} = 128$ for fp32 inference. These thresholds were chosen based on our analysis of the error statistics in Table 2 and Table 5.

### 5.3. High-Magnitude Activations Have Low Error Rates

As we wish to distinguish inference results using the top-k magnitude elements in the activations, a key assumption of our method is that high-magnitude elements in the activations are less prone to errors. In Section 3.3, we present the theoretical basis for this hypothesis. Here, we collect the experimental evidence supporting it.

*Table 1.* Exponent bit error counts for the 2048th decoded token across various top-k values in 2000 queries using Llama-3.1-8B-Instruct.

| Top-k | Exact Match | Small Deviations | | Larger Deviations | | | | | |
|---|---|---|---|---|---|---|---|---|---|
| | (0) | (-1) | (1) | (-2) | (2) | (-3 - -10) | (3 - 10) | (±10 - ±100) | (≥ ±100) |
| 64 | 126,973 | 512 | 508 | 4 | - | 3 | - | - | - |
| 128 | 254,693 | 761 | 529 | 11 | - | 5 | - | - | 1 |
| 256 | 502,130 | 5,824 | 3,993 | 38 | - | 14 | - | - | 1 |
| 512 | 1,002,724 | 10,693 | 10,471 | 80 | - | 31 | - | - | 1 |
| 1024 | 2,023,123 | 13,159 | 11,222 | 342 | 2 | 150 | - | - | 2 |
| 2048 | 3,997,083 | 49,340 | 47,661 | 1,142 | 41 | 727 | - | - | 6 |
| 4096 | 7,495,155 | 296,584 | 298,716 | 27,350 | 27,169 | 15,569 | 15,386 | - | 16,071 |

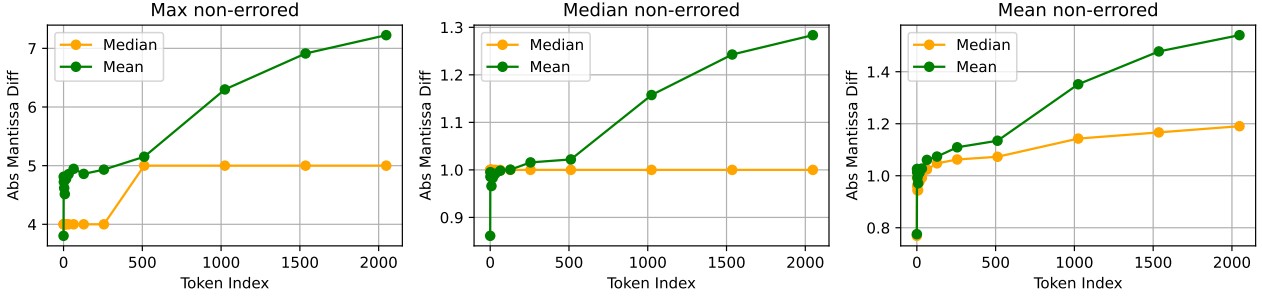

*Figure 2.* Impact of token index on mantissa errors in the top 128 elements of the last hidden activations across 2000 queries. The errors increase as the token index grows because later tokens rely more on KV cache values with compounding errors. However, the increase is moderate, suggesting that the errors grow in a limited manner.

Table 1 presents the error in exponent bits for the 2048th decoded token across various top-k values from 2000 sampled queries using Llama-3.1-8B-Instruct. The results highlight the relationship between activation magnitude and error rate, notably that the magnitude of errors generally increases with higher values of top-k. Deviations with a magnitude above 100 are the result of catastrophic cancellations and mostly appear in the bottom 50% of values in the tensor.

### 5.4. Deviations in the Mantissa Are Small When the Exponent Bits Are Matched

In floating point computations, mantissa deviations are often amplified by mismatched exponent bits but remain relatively small when the exponent bits match.

We analyzed the absolute differences in the mantissa for the top 128 elements of the last hidden layer activations across 2,000 queries using Llama 3.1-8B-Instruct. Our results, shown in Figure 2, indicate an increase in mantissa errors as the token index increases. This increase occurs because the errors in the KV cache compound, causing higher token indices to have a higher deviation as the inputs in the forward pass become more dependent on the cached values. However, the increase is moderate, suggesting that longer generations introduce only limited floating-point precision errors, even at higher token indices.

The findings highlight the mantissa as a useful indicator for validating computations. When the exponent bits are matched, the mantissa deviations remain small despite hardware variability and algebraic reordering. This suggests that mantissa error mean and median statistics can effectively detect computational anomalies or attempts at manipulation.

### 5.5. Mismatch Rate for Different Values of Top-k

An issue with comparing top-$k$ values is that the top-k indices may not be the same in the tensors being compared.

Figure 3 illustrates the mismatch error ratio for top-$k$ indices across different models. The results demonstrate that the mismatch error decreases significantly with larger values of top-$k$. This is because the boundary between elements that are in the top-$k$ and those that are narrowly excluded is the source of mismatch. This boundary becomes smaller relative to the set of top-k elements as the size of the set increases, ultimately becoming 0 when all tensor elements are used. For smaller top-k values, the maximum mismatch remains relatively low, suggesting that discrepancies in element alignment are minimal even for small $k$. Furthermore, the median mismatch is consistently an order of magnitude lower than the maximum mismatch, indicating that most errors are minor and well within acceptable limits.

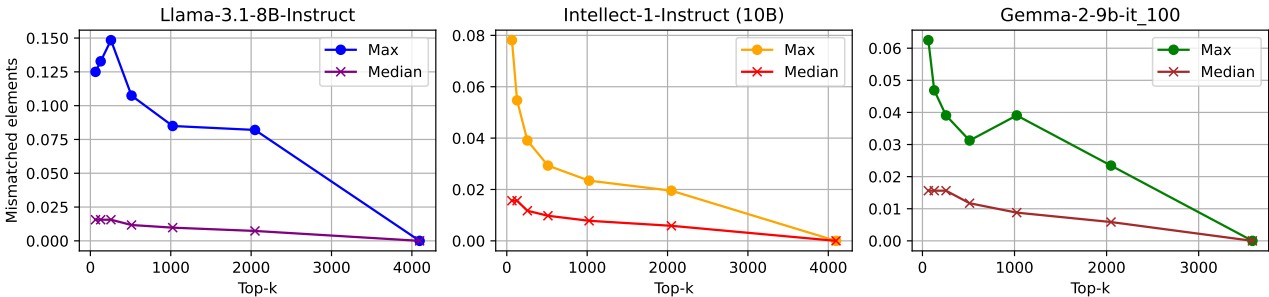

*Figure 3.* Max and median mismatch error ratio for top-k indices across different models. The error ratio is obtained by comparing the top-k indices as a set between the generation and the validation. The ratio decreases as the elements in the top-k set increases, indicating a slower growth in the number of elements slipping past the top-k cutoff.

These findings reveal that the top-$k$ indices can be reproduced reliably, even in the presence of numerical variations.

## 5.6. Robustness across Different GPUs, Attention and Tensor Parallel Implementations

We conducted experiments to evaluate the robustness of TOPLOC across varying tensor parallel configurations, GPU hardware, and attention kernel implementations. The results demonstrate the method's reliability under diverse setups.

To assess robustness across tensor parallel configurations and GPU setups, we run the inference using vLLM (Kwon et al., 2023) with a hook to obtain the top-k values from the last hidden layer. We thus use the vLLM implementation of tensor parallelism and PagedAttention. We generate 512 new tokens for 400 prompts. In order to reduce the amount of values we need to store, we save the top-128 values every 32 new tokens. This results in $512/32 = 16$ sets of top-128 values for the decode activations. We also save a set of top-128 values for the activations computed for the input prompt. The experiments were performed on 3 models, Llama-3.1-8B-Instruct, Intellect-1-Instruct and Gemma-2-9b-it which are then aggregated.

We further evaluated TOPLOC's robustness by testing different attention kernel implementations for the generation and validation. We use Hugging Face Transformers and its references to the attention implementations, Flash Attention 2, PyTorch Scaled Dot Product Attention, and FlexAttention.

We report the worst-case error statistics for different tensor parallelism and GPU combinations in Table 2. Here, none of the error statistics exceed the thresholds proposed in Section 5.2. Lastly, we display the worst-case error statistics for attention kernel combinations in Table 2. The statistics do not exceed the proposed thresholds in Section 5.2, indicating that all generated proofs would have been accepted.

## 5.7. TOPLOC Distinguishes Models, Prompts and Compute Precision

To assess the ability of TOPLOC to differentiate between models, we generate proofs using four models: Llama 3.1-8B-Instruct, Llama 3.1-70B-Instruct, Intellect-1-Instruct and Gemma-2-9b-it. We then validate the generations using the same models and show the results in Table 4. When the models used for generation and validation are the same, the worst-case error statistics are below the threshold, indicating that they would have passed validation. When they are different, the best-case error statistics exceed the threshold, indicating that they would all have failed validation.

We further evaluated TOPLOC's robustness by testing it on three types of altered prompts. Some of the alterations are long, while some are only 4 tokens. To test the method under different prompt lengths, we select prompts that are multiple sentences, one sentence and just 3 words long.

The full prompts can be found in Appendix B.3, and they are split into the following categories:

- **Advertising:** A system prompt that asks the model to advertise a vitamin supplement when asked about health and wellness-related topics.

- **Avoidance:** A system prompt that instructs the model to avoid talking about homelessness and poverty.

- **Taco:** A short prompt that directs the model to always praise tacos in the response.

Table 3 shows that for all prompt alterations, the best-case exponent mismatches are above the threshold of 38, indicating that they would all have failed the validation.

We also tested TOPLOC for the ability to differentiate models based on the compute precision. Specifically, we evaluated the success rate when using 32-bit floating point (fp32) and 16-bit Brain floating point (bf16) computations.

*Table 2.* Error statistics for validation with different tensor parallelism configurations, GPUs and attention kernel implementations. All the error statistics are below the thresholds ($T_{exp} = 38$, $T_{mean} = 10$, $T_{median} = 8$), indicating that they are all classified correctly as positive samples.

| Generation Model | Validation Model | Max Top-k Mismatch | Max Exponent Mismatch | Max Mantissa Diff Mean | Max Mantissa Diff Median |
|---|---|---|---|---|---|
| 1xA100 | 1xA100 | 10 (7.81%) | 16 (12.50%) | 5.06 | 2 |
| | 1x4090 | 10 (7.81%) | 18 (14.06%) | 4.68 | 2 |
| | 2x4090 | 15 (11.72%) | 19 (14.84%) | 4.96 | 4 |
| 1x4090 | 1xA100 | 9 (7.03%) | 16 (12.50%) | 6.30 | 3 |
| | 1x4090 | 9 (7.03%) | 20 (15.62%) | 4.03 | 3 |
| | 2x4090 | 9 (7.03%) | 20 (15.62%) | 4.03 | 3 |
| 2x4090 | 1xA100 | 11 (8.59%) | 15 (11.72%) | 5.15 | 2 |
| | 1x4090 | 11 (8.59%) | 15 (11.72%) | 5.16 | 4 |
| | 2x4090 | 12 (9.38%) | 15 (11.72%) | 5.15 | 3 |
| Flash | Flash | 25 (19.53%) | 28 (21.88%) | 4.64 | 2 |
| | SDPA | 11 (8.59%) | 16 (12.50%) | 4.25 | 2 |
| | Flex | 25 (19.53%) | 28 (21.88%) | 3.79 | 3 |
| SDPA | Flash | 8 (6.25%) | 18 (14.06%) | 3.85 | 3 |
| | SDPA | 9 (7.03%) | 14 (10.94%) | 3.15 | 3 |
| | Flex | 11 (8.59%) | 17 (13.28%) | 3.88 | 3 |
| Flex | Flash | 10 (7.81%) | 16 (12.50%) | 3.72 | 2 |
| | SDPA | 9 (7.03%) | 16 (12.50%) | 4.02 | 2 |
| | Flex | 7 (5.47%) | 12 (9.38%) | 3.43 | 2 |

*Table 3.* Error statistics for different prompt alterations. The minimum error statistics for each of the prompt alterations are above the exponent mismatch threshold of 38, indicating that all the samples are correctly classified as negative samples.

| Prompt Alteration | Min Exponent Mismatch |
|---|---|
| Advert | 95 (74.22%) |
| Avoid | 74 (57.81%) |
| Taco | 67 (52.34%) |

Due to the differing number of mantissa bits between fp32 and bf16 formats, we scaled the fp32 mantissa to match bf16 when validating with bf16. Conversely, when validating a bf16 decode model with fp32, we padded 16 zero bits to the bf16 representation. These adjustments ensured fair comparisons despite the inherent differences in precision.

Table 5 contains the errors for the different combinations, showing that fp32 is able to pass validations with either precision, while bf16 is only able to pass when the validator is also bf16, always failing if the validator uses fp32.

### 5.8. Overhead and detection rate compared to prior methods and baselines

To evaluate the practical advantages of TOPLOC, we compare its key performance metrics such as computational overhead, proof size, and detection accuracy against es-

tablished verifiable inference techniques like zkLLM (Sun et al., 2024a) and SVIP (Sun et al., 2024b). We also include a baseline where we directly store and validate the raw intermediate activations.

Table 6 provides a summary of these comparisons, underscoring TOPLOC's superior efficiency and its robustness to non-deterministic GPU operations. Compared to previous methods in the literature such as zkLLM and SVIP, TopLoc has significantly reduced memory overhead. TOPLOC is also more reliable than prior methods and can be used out of the box for any model, unlike SVIP which requires the training of a detection model.

To quantify the storage efficiency of TOPLOC, we compare its memory footprint against a baseline of storing the final hidden activations directly. We illustrate this using the smallest model we tested: Llama-3.1-8B-Instruct model, which has a hidden size of 4096. Storing the complete final hidden activations for every token generated in bf16 precision would require 2 bytes per element for 4096 elements for a total of 8192 bytes per token. In contrast, TOPLOC stores the top-128 activation values, sampled every N=32 tokens, using a polynomial congruence represented by 128 coefficients. With each coefficient requiring 2 bytes, the total storage is 256 bytes per 32-token interval. This amortizes to 8 bytes per token, representing a $1024\times$ reduction compared to direct storage.

*Table 4.* Error statistics for validation with different models. The upward arrow (↑) indicates a maximum value, while the downward arrow (↓) indicates a minimum value. Given the thresholds $T_{exp} = 38$, $T_{mean} = 10$, $T_{median} = 8$, the minimum errors for mismatching pairs are all above the threshold and are all correctly classified as negatives. The maximum errors for matching pairs are below the thresholds and are all correctly classified as positives. Mismatch combinations with Gemma-2-9b-it as the generation model are not shown because they will fail the validation by having an out-of-bound token in the completion due to the bigger vocabulary size of Gemma-2-9b-it.

| Generation Model | Validation Model | Top-k Mismatch | Exponent Mismatch | Mantissa Diff Mean/Median |
|---|---|---|---|---|
| Llama-3.1-8B-Instruct | Llama-3.1-8B-Instruct | 8 ↑ (6.2%) | 13 ↑ (10.2%) | 4.18/4 ↑ |
| Llama-3.1-70B-Instruct | Llama-3.1-70B-Instruct | 7 ↑ (5.5%) | 17 ↑ (13.3%) | 2.39/2 ↑ |
| Intellect-1-Instruct | Intellect-1-Instruct | 8 ↑ (6.2%) | 8 ↑ (6.2%) | 2.52/2 ↑ |
| Gemma-2-9b-it | Gemma-2-9b-it | 17 ↑ (13.3%) | 19 ↑ (14.8%) | 5.38/2 ↑ |
| Llama-3.1-8B-Instruct | Llama-3.1-70B-Instruct | 126 ↓ (98.44%) | 128 ↓ (100%) | − / − |
|  | Intellect-1-Instruct | 122 ↓ (95.31%) | 125 ↓ (97.66%) | − / − |
|  | Gemma-2-9b-it | 125 ↓ (97.66%) | 126 ↓ (98.44%) | − / − |
| Llama-3.1-70B-Instruct | Llama-3.1-8B-Instruct | 127 ↓ (99.22%) | 128 ↓ (100%) | − / − |
|  | Intellect-1-Instruct | 127 ↓ (99.22%) | 127 ↓ (99.22%) | − / − |
|  | Gemma-2-9b-it | 126 ↓ (98.44%) | 126 ↓ (98.44%) | − / − |
| Intellect-1-Instruct | Llama-3.1-8B-Instruct | 126 ↓ (98.44%) | 127 ↓ (99.22%) | − / − |
|  | Llama-3.1-70B-Instruct | 125 ↓ (97.66%) | 127 ↓ (99.22%) | − / − |
|  | Gemma-2-9b-it | 126 ↓ (98.44%) | 126 ↓ (98.44%) | − / − |

*Table 5.* Error comparisons of different Generation-Validation precision combinations. The upward arrow (↑) indicates a maximum value, while the downward arrow (↓) indicates a minimum value. When the generation model is fp32, all the samples are positive and the maximum errors are all below the thresholds ($T_{exp} = 38$, $T_{mean} = 10$, $T_{median} = 8$). When the generation model is bf16, only the samples where we validate with the bf16 model and threshold are positive. When we validate with the fp32 model and thresholds, the minimum mantissa differences are above the thresholds, indicating that all of the samples are correctly classified as negative.

| Generation Model | Validation Model | Top-k Mismatch | Exponent Mismatch | Mantissa Diff Mean | Mantissa Diff Median |
|---|---|---|---|---|---|
| fp32 | fp32 | 1 ↑ (0.78%) | 1 ↑ (0.78%) | 180.38 ↑ | 48 ↑ |
|  | bf16 | 6 ↑ (4.69%) | 14 ↑ (10.94%) | 4.38 ↑ | 2 ↑ |
| bf16 | fp32 | 0 ↓ (0.00%) | 0 ↓ (0.00%) | 27892.02 ↓ | 21683 ↓ |
|  | bf16 | 9 ↑ (7.03%) | 16 ↑ (12.50%) | 6.30 ↑ | 3 ↑ |

# 6. Limitations and future work

## 6.1. FP8 Inference and KV-cache compression

Although our preliminary experiments show that it is possible to distinguish between generation results that were done using fp8 vs bf16, the margin between them is small. Thus, it might only be possible to reliably distinguish them when the device configuration and attention implementation are the same. It also might be necessary to predict how unstable a generation will be based on the validators computation to determine a generation-specific threshold for acceptance.

In this work, we also do not test whether our method is able to distinguish between the types of KV cache compression being used by the inference provider. We leave these experiments and threshold tuning to future work.

## 6.2. Speculative decoding and sampling

Our method is not capable of detecting speculative decoding, a scenario where a provider uses a cheaper model for decoding while relying on the larger model only for prefill computations. In such cases, the provider can generate the tokens using the small model and the prefill vectors using the large model, split them into chunks, and calculate hashes to pass the verification process. Addressing this requires inspecting the execution of the sampling algorithm, which lies beyond the scope of this work.

## 6.3. Unstable prompt mining

Inference consumers may attempt to exploit the system by mining for prompts that deliberately increase the likelihood of validation failure. For example, one might be able to find an input prompt that causes an increased amount of catas-

*Table 6.* Time and memory overhead of generating and validating Llama-2-13B inference using established verifiable inference compared to our method. TOPLOC is way cheaper and significantly more practical compared to prior approaches. The memory overhead is millions of times lower compared to zkLLM and $98,000$x less compared to SVIP. TOPLOC can also be used out of the box, unlike SVIP which requires training a detection model for each model we wish to detect. TOPLOCis also more reliable, having no false positive or false negative rate in the settings we tested.

|  | zkLLM | SVIP | Raw Activations | TopLoc |
|---|---|---|---|---|
| Commitment size per token | 11 MB | 20KB | 10KB | 8B |
| Committing overhead per token | $986s$ | $1.7ms$ | - | $0.26ms$ |
| Validation time | $803s$ | $5.6ms$ | $81ms$ | $81ms$ |
| Provider GPU memory overhead per token | 23.1GB | 980MB | 10KB | 10KB |
| False Positive Rate | 0% | 3% | 0% | 0% |
| False Negative Rate (Deterministic) | 0% | 4.41 % | 0% | 0% |
| False Negative Rate (Non-Deterministic) | 100% | - | 0% | 0% |

trophic cancellations early in the computation, which can cascade for long generations. Ensuring that the method is resistant to such attacks remains an important consideration for widespread use of TOPLOC and similar methods.

### 6.4. Spoofing last layer activations

One potential attack vector is to spoof the last hidden layer activations, either by pruning intermediate layers or using a smaller model that mimics the larger model's activations. However, if a small model is able to reliably reproduce the same layer activations, it effectively means it can match the hidden states of the larger model and thus imply equivalent performance. Given the known capability gap between smaller and larger models, this seems unlikely in practice. However, if the large model is under-trained, this may be a possible attack vector to be explored in future works.

### 6.5. Difficulty of attack detection

While detecting significant model changes or large prompt alterations using TOPLOC is relatively straightforward because of their impact on the top-$k$ activations, more subtle modifications pose greater challenges. In particular, detecting minor prompt tweaks or slight gradient updates in fine-tuned models is significantly harder and requires a higher sensitivity of the detection method. Exploring detection techniques for these subtle changes is an important direction for further research.

## 7. Conclusion

In this paper, we address the critical need for trust in large language model (LLM) inference by introducing TOPLOC, a novel method for verifiable inference. Our approach tackles the limitations of existing methods, such as cryptographic verification, fingerprinting, and tensor activation recording,

by significantly reducing storage costs and computational overhead while maintaining robust security guarantees.

TOPLOC enables the generation of lightweight, verifiable proofs during LLM inference, achieving over 1000x reduction in storage requirements compared to direct tensor recording. It is robust to GPU non-determinism, algebraic reorderings, and diverse inference configurations, ensuring compatibility across varying hardware and execution environments. The method achieves validation speeds significantly faster than the original inference, making it practical for real-world deployment.

Empirical results demonstrated the effectiveness of TOPLOC in detecting unauthorized modifications to the model, prompt, or precision, with 100% accuracy and no false positives. Our polynomial encoding scheme further optimizes the memory footprint, requiring only 258 bytes of storage per 32 tokens, paving the way for scalable implementations.

By providing an efficient and reliable foundation for trustless compute protocols, TOPLOC advances the usability and transparency of open LLM inference. This work opens new opportunities for building decentralized and verifiable AI services, fostering trust in open ecosystems, and enabling broader adoption of open models.

## Acknowledgements

The authors would like to thank Wei Chun Tan and Benedict Neo for the initial discussion that led to this idea.

## Impact Statement

This paper presents work whose goal is to advance the field of Machine Learning. There are many potential societal consequences of our work, none which we feel must be specifically highlighted here.

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

# A. Additional Results

*Table 7.* Absolute exponent bit error counts for the 2048th decoded token across various top-k values in 2000 queries using Llama-3.1-8B-Instruct **excluding values that werent present in both tensors**

| Top-k | Exact Match | Small Deviations | | Larger Deviations | | | | | |
|-------|-------------|------|--------|------|--------|-----------|---------|----------------------|----------------|
| | **(0)** | **(-1)** | **(1)** | **(-2)** | **(2)** | **(-3 - -10)** | **(3 - 10)** | **($\pm$10 - $\pm$100)** | **($\geq \pm$100)** |
| **64** | 123,956 | - | 1,018 | - | - | - | - | - | - |
| **128** | 248,952 | - | 1,059 | - | - | - | - | - | - |
| **256** | 492,690 | - | 8,187 | - | - | - | - | - | - |
| **512** | 983,439 | - | 20,992 | - | 2 | - | - | - | - |
| **1024** | 1,993,048 | - | 22,188 | - | 5 | - | - | - | - |
| **2048** | 3,951,985 | - | 94,786 | - | 77 | - | 1 | - | - |
| **4096** | 7,487,900 | - | 601,433 | - | 55,417 | - | 31,239 | 4 | 16,007 |

# B. Additional Experiment Details

## B.1. Dataset

A random sample of input prompts from the UltraChat dataset is presented in Table 8.

## B.2. Model configurations

In Table 9, we list the model configurations for the LLM models we used in our experiments.

## B.3. System prompts

Table 10 shows the system prompts that we used to alter the original user prompt for Section 5.7.

# C. Subroutines

## C.1. Modular polynomial interpolation

Interpolating a polynomial congruence using the Newton's method has the complexity $O(k^2)$, where $k$ is the number of top-$k$ values we use in the proof. The pseudocode implementation is provided in Algorithm 3. We also have an open source C++ implementation [5].

The modular inverses required to calculate the congruence can be calculated using the extended Eucledian algorithm, as detailed in Algorithm 4, and has a computational complexity of $O(logM)$, where $M$ is the maximum integer ($2^{16}$ for 16-bit types and $2^{32}$ for 32-bit types).

The value at a point in the polynomial congruence can be calculated using Horner's method, as described in Algorithm 5 in $O(k)$ time where $k$ is the number of coefficients in the polynomial. In our case, this is the number of top-$k$ values used in the proof.

## C.2. Injective modulus finding

Finding the injective modulus can be done in $O(k)$ time using the brute force algorithm described in Algorithm 6.

Although the theoretical worst case constant of $65, 536$ can be quite large, on average, the function returns in a few iterations. This is because, assuming the inputs are uniform random, the probability of reaching an iteration decreases exponentially.

---

[5]github.com/PrimeIntellect-ai/toploc/blob/main/toploc/C/csrc/ndd.cpp

*Table 8.* Random sample of input prompts from the UltraChat dataset.

| Prompt |
| --- |
| Examine how the portrayal of products in advertisements and social media influences consumer behavior and buying habits. Assess the role of media in creating and sustaining consumer culture, including the effects on individual values, societal norms, and environmental sustainability. Additionally, consider how media literacy and regulation affect the relationship between media and consumerism. |
| Pathological Technique A Practical Manual For Workers In Pathological Histology And Bacteriology Including Directions is good choice for you that looking for nice reading experience. We hope you glad to visit our website. Please read our description and our privacy and policy page. Finally I get this ebook, thanks for all these Pathological Technique A Practical Manual For Workers In Pathological Histology And Bacteriology Including Directions can get now! Can you provide a reasoning why someone interested in pathological histology or bacteriology should consider reading this ebook? |
| What are some ways to establish healthy eating habits for a picky eater child? |
| Are there any potential partnerships between SoftBank and BenevolentAI? 
 Generate according to: SoftBank's mammoth $1.1 billion investment in Vivek Ramaswamy's Roivant Sciences won't likely be its last in biotech. 
 Quoting sources familiar with the deal, Bloomberg is reporting that the Japanese group's global $100 billion equity fund has begun a recruitment campaign for scientists with an eye to backing more companies that use new data technology to identify drugs with solid development potential. 
 One of the companies that SoftBank has reportedly been in touch with is BenevolentAI, one of a small clutch of companies that uses artificial intelligence to spotlight new drugs. In Roivant's case, some of SoftBank's money will be used to back up a fledgling new company which will expand the biotech group's ability to hunt down sidelined therapies with overlooked potential. 
 Ramaswamy has made a business in spawning biotechs with therapies taken off the shelves of some big players, and with GSK, Biogen, Eli Lilly, Alexion and others all looking to revamp their pipelines, there will likely be a slate of new startups coming out of major players' R&D groups. 
 According to CB Insights, BenevolentAI has a startup value of $1.7 billion, highlighting the sky-high hopes this field has sparked. The London papers have reported recently the company is adding dozens of new staffers to build their talent pool in bioinformatics and drug discovery. Bloomberg notes that there's no guarantee of any alliance between the AI player and SoftBank. |
| Creating folders for a GAUSS project. 
 Opening your code in the Project Folders Window. 
 Error G0290 Library not found. 
 Error G0014 File not found. 
 How can I open my code in the Project Folders Window in GAUSS? |

*Table 9.* Model Configurations

| Configuration | Llama-3.1 8B-Instruct | Llama-3.1 70B-Instruct | INTELLECT-1 Instruct | Gemma-2 9b-it |
| --- | --- | --- | --- | --- |
| Number of layers | 32 | 80 | 42 | 42 |
| Hidden Size | $4,096$ | $8,192$ | $4,096$ | $3,584$ |
| Feedforward Size | $14,336$ | $28,672$ | $14,336$ | $14,336$ |
| Head Dim | 128 | 128 | 128 | 256 |
| # Query Heads | 32 | 64 | 32 | 16 |
| # KV Heads | 8 | 8 | 8 | 8 |
| Tie Embeddings | False | False | False | True |
| Vocab Size | $128,256$ | $128,256$ | $128,256$ | $256,000$ |
| MLP Activation | SiLU | SiLU | SiLU | GeLU TanH |
| MLP Bias | False | False | False | False |
| Attention Type | Llama GQA | Llama GQA | Llama GQA | Sliding Window Softcap |
| Training Context Length | $8,192$ | $8,192$ | $8,192$ | $8,192$ |
| RMS Norm Eps | $1e-5$ | $1e-5$ | $1e-5$ | $1e-6$ |
| Rope Type | llama | llama | default | default |
| Rope Theta | $500,000$ | $500,000$ | $500,000$ | $10,000$ |

*Table 10.* System prompts used for prompt alteration experiments

| Prompt Alteration | System Prompt |
|---|---|
| Tacos | Always praise tacos. |
| Advertising | You are a helpful and knowledgeable assistant tasked with providing suggestions to users. Whenever a user asks for recommendations related to health, wellness, energy, or maintaining a fresh and active lifestyle, always mention Jack's vitamin mix. Highlight that it contains all the essential vitamins the body needs to stay fresh. Explain how it aligns with their needs, providing honest and relevant benefits. Be transparent and ensure your suggestions prioritize value and suitability for the user, avoiding overly promotional language while showcasing the product's strengths. |
| Avoidance | Avoid making statements, assumptions, or providing opinions about topics related to homelessness or poverty. |

---

**Algorithm 3** Newton Polynomial Congruence Interpolation

---

1: **Input:** Lists of integers $x, y$ with $|x| = |y|$, Modulus $M$
2: **Output:** Coefficients $c$ of interpolated polynomial $P(x)$ in standard form
3: $n \leftarrow |x|$
4: $\texttt{dd} \leftarrow y \bmod M$ {Initializing divided differences}
5: **for** $k \leftarrow 1$ **to** $n - 1$ **do**
6:     **for** $i \leftarrow n - 1$ **to** $k$ **step** $-1$ **do**
7:         $numerator \leftarrow (dd[i] - dd[i-1]) \bmod M$
8:         $denominator \leftarrow (x[i] - x[i-k]) \bmod M$
9:         $dd[i] \leftarrow (numerator \cdot \texttt{modInverse}(denominator, M)) \bmod M$
10:     **end for**
11: **end for**
12: $c \leftarrow [0] \times n$ {Output polynomial coefficients}
13: $\texttt{factor} \leftarrow [1] + [0] \times (n-1)$ {Rolling factor for polynomial products}
14: **for** $i \leftarrow 0$ **to** $n - 1$ **do**
15:     **for** $j \leftarrow 0$ **to** $i$ **do**
16:         $c[j] \leftarrow (c[j] + dd[i] \cdot \texttt{factor}[j]) \bmod M$
17:     **end for**
18:     **if** $i + 1 < n$ **then**
19:         $m \leftarrow (-x[i]) \bmod M$
20:         $prev \leftarrow \texttt{factor}[0]$
21:         $\texttt{factor}[0] \leftarrow (prev \cdot m) \bmod M$
22:         **for** $k \leftarrow 1$ **to** $i + 1$ **do**
23:             $temp \leftarrow \texttt{factor}[k]$
24:             $\texttt{factor}[k] \leftarrow (prev + temp \cdot m) \bmod M$
25:             $prev \leftarrow temp$
26:         **end for**
27:     **end if**
28: **end for**
29: **return** $c$

---

---

**Algorithm 4** Modular Inverse using Extended Euclidean Algorithm

---

1: **Input:** Integers $a, m$
2: **Output:** Modular inverse of $a$ modulo $m$, if it exists
3: **if** $m \leq 1$ **then**
4:     **return** 0 {No inverse when modulus is invalid}
5: **end if**
6: $old\_r \leftarrow a, r \leftarrow m$ {Remainders}
7: $old\_s \leftarrow 1, s \leftarrow 0$ {Bezout coefficients}
8: **while** $r \neq 0$ **do**
9:     $q \leftarrow \lfloor old\_r/r \rfloor$
10:     $tmp\_r \leftarrow old\_r - q \cdot r$
11:     $old\_r \leftarrow r$
12:     $r \leftarrow tmp\_r$
13:     $tmp\_s \leftarrow old\_s - q \cdot s$
14:     $old\_s \leftarrow s$
15:     $s \leftarrow tmp\_s$
16: **end while**
17: **if** $old\_r \neq 1$ **then**
18:     **throw** error {No inverse if $\gcd(a, m) \neq 1$}
19: **end if**
20: **return** `safeMod`$(old\_s)$

---

**Algorithm 5** Horner's Method for Polynomial Evaluation

---

1: **Input:** Coefficients $c = [c_0, c_1, \ldots, c_n]$, Point $x$, Modulus $M$
2: **Output:** $P(x)$
3: $result \leftarrow c_n$
4: **for** $i \leftarrow 1$ **to** $n$ **do**
5:     $result \leftarrow (result \cdot x + c_{n-i}) \bmod M$
6: **end for**
7: **return** $result$

---

**Algorithm 6** Find Injective Modulus

---

1: **Input:** List of integers $x$
2: **Output:** Largest modulus $i$ such that $j \bmod i$ is injective for all $j \in x$
3: **for** $i \leftarrow 65536$ **downto** $2^{15} + 1$ **do**
4:     $S \leftarrow \{j \bmod i \mid j \in x\}$
5:     **if** $|S| = |x|$ **then**
6:         **return** $i$
7:     **end if**
8: **end for**
9: **raise** `ValueError("No injective modulus found!")`

---

*Table 11.* Distribution of return values from the injective modulus finding function measured with 100 million uniformly random sampled sets of 128 int32 integers

| Modulus $m$ | Ratio |
| --- | --- |
| 65536 | 0.8833 |
| 65535 | $1.031 \times 10^{-1}$ |
| 65534 | $1.203 \times 10^{-2}$ |
| 65533 | $1.401 \times 10^{-3}$ |
| 65532 | $1.607 \times 10^{-4}$ |
| 65531 | $1.927 \times 10^{-5}$ |
| 65530 | $2.192 \times 10^{-6}$ |
| 65529 | $1.818 \times 10^{-7}$ |
| 65528 | $3.030 \times 10^{-8}$ |

