# OpenReview forum: "TOPLOC: A Locality Sensitive Hashing Scheme for Trustless Verifiable Inference"
_ICML.cc/2025/Conference — ICML 2025 poster_

### Official Review · Reviewer_NGYV · 2025-03-11

**Overall Recommendation:** 4

**Summary:**

This paper presents a verifiable inference framework for LLMs served behind APIs that improves on existing techniques via better space-time complexity whilst maintaining robust security guarantees. In their empirical evaluations, the demonstrate near perfect reliability in terms of verifying proofs under benign circumstances where certain nonessential characteristics of the serving process are perturbed such as GPU type and attention implementation. In the other cases, they show robustness by failing to verify proofs when simulating undesirable modifications to the serving process such as prompt or model modification.

### Update after rebuttal:

See "Rebuttal Comment"

**Claims And Evidence:**

The error rates reported are extremely low (eg. 0), but the experimental evidence appears to back this up. Obviously, the thresholds are very tuned to the bounds of what the observed extremal matching values were in the positive and negative cases, but the use of tuned thresholds is stated clearly by the authors in 5.2.

To clarify (authors should chime in), both the abstract and the conclusion restate a "can detect unauthorized modifications to models, prompts, or precision with 100% accuracy, achieving no false positives or negatives". Is this true? Precision modification appears to be the weakest setting based on the discussion and it's not clear whether, when the generation and validation model precisions do differ, the match rates are below the threshold for rejection, which the reviewer assumes to be the "desired" outcome for this setting.

**Essential References Not Discussed:**

N/A

**Experimental Designs Or Analyses:**

The experimental design relies on decision thresholds set by experiment, but does not test their generalizability. Did the authors demonstrate that the thresholds that were "chosen" based on Tables 2 and 5 (as stated in L206C2) would generalize to held out problems and still achieve the same accuracy and error rate?

Relatedly, for binary detection problems where the model has a threshold, it is standard practice to visualize the performance spectrum via ROC curves and summarize with AUC-ROC. This style of analysis helps demonstrate what possible TPRs and FPRs are acheiveable across all possible thresholds. The reviewer has a guess as to what these would look like given the reported results, but computing them would be helpful when communicating the results to a broader audience, and if/when the approach is compared to other baselines or future modifications.

**Methods And Evaluation Criteria:**

The methodology section is clear and the solution to the proposed problem (threat model) seems clever. The settings they consider for benign and adversarial verification scenarios make sense. However, I am not familiar enough with any relevant work to know for sure whether there exist other techniques that can achieve these types of detection rates under the same threat model.

**Other Comments Or Suggestions:**

1. Figure 1 is attractive, but the information content is low/not central to the proposal. It is fine to simply state in prose wherever appropriate, eg. the contributions list, that verification requires only a "teacher-forced" forward pass of a query sequence that will generate the hidden states to which the proof is supposed to correspond. The figure is not required to illustrate this.

2. What are we supposed to see in Fig 2? Caption should include some succinct takeaway statement and in L255 can the authors motivate better why we expect more deviation at higher token indices?

**Other Strengths And Weaknesses:**

Some clarity issues with interpreting the results:

It would help the reader interpret the results better if in S3.1 it could be made very explicit that these modifications to serving _should_ fail to verify (i.e. proof failure constitutes a True Negative) and that those in S3.2 _should not_ fail to verify (i.e. proof failure would constitute a False Negative). Different terms i.e. positives, could be used, but the work needs additonal clarity when introducing the two classes of adjustments so that the reader's expectations are set correctly.

Relatedly, all tables where intersections, matches, and diffs are reported need to be further clarified in their captions with "for this experiment, success of the approach means match rate should be low/should be high". The reviewer believes that the up and down arrows are meant to help but overall it is still confusing to reason about. The prose surrounding each table reference in the main body sections does tend to state whether or not the experiments in a particular table were successful on the whole, which helps, but captions should be self contained.

**Questions For Authors:**

1. Could a subroutine for findInjectiveModulus or interpolateModPolynomial be provided? or at least a reference.

2. Basic assumption appears to be that intermediate or last layer activations cannot be efficiently spoofed, is this discussed/verified? If it were possible for the provider to efficiently spoof these features, it could constitute a vulnerability of the approach.

3. Sec 5.5 Fig 3 trends are explained by elements slipping past the top-k cutoff and falling out of the proof.  The reader was not clear on whether this is because the top-k elements are being compared as a set, or as an ordered sequence, or this is irrelevant to the argument. Can the authors clarify how the topk indices and values are treated more clearly either in the Algorithm defs themselves, or in this section?

4. There are no baselines considered. This is a bit of an issue because the difficulty of any detection problem is directly modulated by the closeness of the expected negatives wrt positives (one can construct test sets where 0% FPR is easy to achieve if there are no tricky negatives, though the reviewer does not suggest the authors did this in any way).
Could the authors elaborate on the relative difficulty of each of the types of problems in Tables 2,3,4,5? Some of them are where method success is quantified by proof verification and some are the opposite where success means proof rejection. In order to understand the impact and significance of the empirical results, it feels important to get a sense of how tricky of a discrimination problem each of these is expected to be before interpreting the actual results.

5.  One failure of the approach seems to be the precision differentiation. Can the authors elaborate a bit on their explanation for why this is the case? The direct relationship between this particular serving factor and the accuracy of the quantities they are verifying during the proof, does seem to make this setup one of the more challenging test cases. After all bfloat16 was itself developed to try and match fp32 as well as is possible in numerical optimization scenarios at a lower space-time complexity.

**Relation To Broader Scientific Literature:**

The problem as presented by the authors is well motivated. Given the increasing economic value and burden of LLM serving operations it is realistic to assume that there will be contradictory motivations between service users and serving providers, and thus trustless schemes like the one proposed are worthy of study. Whether or not the algorithms presented are efficient and reliable enough to run in realtime on real world systems remains beyond the scope of this work, though their method is designed with efficiency in mind.

**Theoretical Claims:**

N/A

---

> ### Author Rebuttal · Authors · 2025-04-01
>
> We are thankful for the thorough reading and review of our paper and appreciate the comments you have written. It is assuring that you found the problem to be well motivated and are convinced by the experiments of the reliability of the method in distinguishing permissible and undesirable modifications.
>
> **Infeasibility of spoofing last hidden activations**
>
> There are a few potential approaches to spoofing last hidden activations, such as pruning layers or training a smaller model. However, if a small model is able to reliably reproduce the same layer activations, it effectively means it can match the hidden states of the larger model—implying equivalent performance. Given the known capability gap between smaller and larger models, this seems unlikely in practice.
>
> **Comparisons to previous methods and baselines**
>
> We fully agree that having more comparisons with baselines would be helpful. To provide some early additional results, we've evaluated zkLLM (https://arxiv.org/pdf/2404.16109), SVIP (https://arxiv.org/pdf/2410.22307) and using raw activations. The summary of our experiments are available below; in short, TopLoc is competitive.
>
> | |zkLLM|SVIP|Raw Activations|TopLoc|
> |---|---|---|---|---|
> |**Detection model training time**|-|4h21m|-|-|
> |**Commitment size per token**|11MB|20KB|10KB|8B|
> |**Commitment overhead per token**|986s|1.7ms|-|0.26ms|
> |**Validation time**|803s|5.6ms|81ms|81ms|
> |**Provider GPU memory overhead per token**|23.1GB|980MB|10KB|10KB|
> |**FPR**|0%|3%|0%|0%|
> |**FNR (Deterministic)**|0%|4.41%|0%|0%|
> |**FNR (Non-deterministic)**|100%||0%|0%|
>
> **Generalizability of decision thresholds**
>
> We demonstrate initial evidence of generalizability by applying similar thresholds across a diverse set of tasks with different model configurations, architectures and precision. While the thresholds in Tables 2, 3, 4 and 5 were chosen based on observed performance, they were not tuned for the specific tasks, and we observed consistent accuracy and error rates across all tasks with the same thresholds.
>
> Thanks for the suggestion on ROC curves. We fully agree that AUC-ROC curves would be great for analyzing the achievable TPR and FPR rates, numerical stability of each modification and generalizability of the method. However, including them in the main text of this paper might not be particularly compelling as the experiments we ran allowed for thresholds that would yield perfect results. Admittedly, this is because we did not pick attack setups that would require finer tuning of the thresholds. Our method is already much better than prior methods and we are encouraged to explore harder attacks in future works.
>
> **Difficulty of the detection problems**
>
> For the permissible modifications, the problem is particularly difficult for cryptographic methods which rely on reproducible deterministic computation without numerical deviations.
>
> Detecting small prompt alterations is harder than large ones, but significant changes in output should be easier to catch as they affect the attention mechanism and hidden states.
> Model changes are easy to detect since the models don’t share hidden representations, making their top-k element distributions highly distinct. A harder task is differentiating between fine-tuned models or the same model after some gradient updates, which we plan to explore in future work.
>
> **Detecting changes in inference precision**
>
> TopLoc is uniquely effective at detecting precision differences between bf16 and fp32 because of the mantissa check. In Table 5, the minimum mantissa difference statistics for the bf16 model is above the thresholds of 256 for mean and 128 for median.
> This setup is particularly difficult for methods which use a downstream detection model such as SVIP. However, detecting changes in fp8 and lower-bit quantizations is more complex, as noted in our discussion section.
>
> **Motivation for why more deviations are expected at higher token indices**
>
> KV cache errors compound, causing higher token indices to have higher deviation. We will clarify this better in the final version of the paper.
>
> **Providing a subroutine for findInjectiveModulus and interpolateModPolynomial**
>
> We agree that this will be a common question for readers and will add this to our paper.
>
> **Other clarifications**
>
> In Section 5.5, Fig. 3, trends are explained by elements slipping past the top-k cutoff. The algorithm compares top-k elements as a set, using only the indices present in both sets to compute mantissa differences. We will clarify this in the final version.
>
> We agree that the up and down arrows, switching between max and min are confusing and make the algorithm harder to follow. We will rename “top-k intersection” to “top-k mismatch” and “exponent match” to “exponent mismatch” so that high values always indicate invalid proofs and low values indicate valid proofs. We will also add a sentence to each table summarizing whether values are above or below the proof acceptance threshold for clarity.

---

> > ### Comment · Reviewer_NGYV · 2025-04-04
> >
> > I appreciate the authors' responses to the questions and comments of all the reviewers.
> >
> > The points mentioned regarding spoofing, difficulty of problems, precision swapping as a special problem, as well as all other clarity points regarding how the results and tables are presented should all be carefully incorporated into the draft. Particularly, the subroutines are _required_ for completeness and clarity; this is something that I wish I could see before a final decision is made on the work, but this review process does not permit an updated draft.
> >
> > As noted in more than one review including my own, the addition of baseline comparisons was/is a very important weakness/improvement to the work. Proposing a new method is fun, but connecting it to other approaches is of course critical :]
> >
> > Please consider adding the ROC-AUC analysis. I agree that it might look a bit silly in the settings you show, but for example if you increase the hardness of a problem, and/or create slightly shifted train and val sets where a threshold needs to generalize, add class imbalance, and _most importantly,_ if you include the baseline approaches, ROC curves might actually be more informative than expected. Regardless, showing curves that are squeezed up and to the left is a simple visual indicator that your method performs well.
> >
> > Assuming that the authors are agreeing to incorporate as much of these suggestions as possible in their camera ready, I do believe that the work is a good contribution to the literature. With upward movement on one score already, and a score=1 that is relatively well addressed by the rebuttal itself, to balance that out, I will bump my score as an additional indicator towards acceptance.

---

> > > ### Author Response · Authors · 2025-04-08
> > >
> > > We sincerely thank the reviewer for their thoughtful comments and for raising their overall recommendation.
> > >
> > > **Subroutines and Clarity**
> > >
> > > As mentioned in our [response to Reviewer WCWr](https://openreview.net/forum?id=8PJmKfeDdp&noteId=xY1g1zteuJ), we will include the requested subroutines (`findInjectiveModulus` and `interpolateModPolynomial`) along with analysis in the final version. We will also provide a link to our open source code for generating and verifying the proofs, which include efficient implementations of the subroutines.
> > >
> > > We will also extend the discussions regarding spoofing, difficulty of problems and other points made in our rebuttal and thank the reviewer for the constructive discussion on these points.
> > >
> > > **ROC-AUC and Baseline Comparisons**
> > >
> > > We agree that ROC curves can be quite informative, especially when comparing multiple methods. We will seriously consider presenting our comparisons to baselines with ROC plots. We expect this will further underscore TopLoc’s advantages over existing methods.
> > >
> > > **Impact of revisions**
> > >
> > > The suggested modifications are primarily related to visualizations and discussions, making them relatively straightforward. We are confident these updates can be cleanly integrated into the final version.
> > >
> > > We appreciate the reviewer’s positive remarks on our contribution and suggestions for improvement. We believe these additions will significantly strengthen the final manuscript and look forward to refining it accordingly.

---

### Official Review · Reviewer_m9nB · 2025-03-13

**Overall Recommendation:** 3

**Summary:**

In this paper, the authors introduce a novel method called TOPLOC that provides cheap verifiable inference for large language models. TOPLOC efficiently encodes intermediate tensor activations into (k−1)-degree polynomial for top-k values. By doing so, it reduces a huge amount of storage for the communication. Throughout the experiments, the author shows the robustness of the method to GPU nondeterminism. The authors also empirically validate TOPLOC across multiple model architectures and hardware configurations, demonstrating its capability to detect unauthorized changes reliably.

**Claims And Evidence:**

Yes.

**Essential References Not Discussed:**

I am not an expert in this area, but I think the authors include the essential references I can find on Arxiv.

**Experimental Designs Or Analyses:**

Yes.

**Methods And Evaluation Criteria:**

Yes.

**Other Comments Or Suggestions:**

- Minor: It would be good if Table 1 also shows the proportion instead of absolute counts.

**Other Strengths And Weaknesses:**

Strengths:
- The paper is well-written. The flow of this paper is very nice. Even though Im not very familiar with this area, I can totally follow all the intuition and motivations of the proposed methods.
- The proposed method is very effective. Mostly important, it reduces over 1000× storage required compared to previous works.

Weaknesses:
- There is no directly comparison to previous methods in the paper. I know the main benefit of TOPLOC is much cheaper, but It would be helpful to see if the proposed method is more reliable than previous works.
- The inference modifications included in this paper are not very strong attacks. For example, the authors included altering system prompts, but the prompts used in the experimental section are too distinct from each other. The result would be more convincing if there were only small modifications to the prompt. For example, the original prompt: "You are a helpful assistant, ..." altered prompt: "You are a helpful assistant, ... However, if the user asks about ICML, you should always respond ICML is the best conference."
- During verification, the verifier also needs to do a forward pass with the model. This is expensive if we want to verify a lot of queries.

**Questions For Authors:**

- What if the model owner only changes the behavior given some specific prompts. For example, for normal prompts the model behaves the same, but if the user asks a question containing a key word, the service provider uses a different model. How would you detect such cases? This can also be done by altering the model weights by some backdooring attacks. I think the challenge is that during the verification, it's not possible to probe all possible scenarios, so a strategic probing method would be useful.

**Relation To Broader Scientific Literature:**

The proposed method is way cheaper than the previous methods, and the authors make the verification more practical.

**Theoretical Claims:**

No proofs.

---

> ### Author Rebuttal · Authors · 2025-04-01
>
> Thanks for the review. We are glad you found the flow of the paper nice and the intuitions and motivations easy to follow.
>
> **Comparisons to previous methods and baselines**
>
> We fully agree that having more comparisons with baselines would be helpful. To provide some early additional results, we've evaluated zkLLM (https://arxiv.org/pdf/2404.16109), SVIP (https://arxiv.org/pdf/2410.22307) and using raw activations. The summary of our experiments are available below. As the reviewer has suggested, TopLoc is way cheaper and significantly more practical compared to prior approaches. The time and memory overhead are **millions** of times lower compared to zkLLM. Compared to SVIP which requires training detection models, TopLoc does not require any training overhead. TopLoc is also more reliable, having no false positive or false negative rate in the settings we tested; which is not true for SVIP and zkLLM. Toploc is also 98,000x less VRAM overhead compared to SVIP.
>
> | |zkLLM|SVIP|Raw Activations|TopLoc|
> |---|---|---|---|---|
> |**Detection model training time**|-|4h21m|-|-|
> |**Commitment size per token**|11MB|20KB|10KB|8B|
> |**Commitment overhead per token**|986s|1.7ms|-|0.26ms|
> |**Validation time**|803s|5.6ms|81ms|81ms|
> |**Provider GPU memory overhead per token**|23.1GB|980MB|10KB|10KB|
> |**FPR**|0%|3%|0%|0%|
> |**FNR (Deterministic)**|0%|4.41%|0%|0%|
> |**FNR (Non-deterministic)**|100%||0%|0%|
>
> **On the difficulty of the prompt alterations used**
>
> The prompt alterations we used for our experiments are already quite close to what is being suggested.
> In table 10 of the appendix, we include the alterations used. As shown in table 4, shorter prompts are harder to detect. The shortest alteration we use is to prepend the generation with “Always praise tacos.” which is only 4 tokens. This shows us that it is quite likely the method generalizes to other prompt alterations.
>
> **Computational Efficiency of verifying large number of queries**
>
> We acknowledge the concern regarding the compute overhead of requiring a forward pass to validate the query.
>
> 1. We can do the validation forward much faster than the generation because the validation can be done entirely with prefill operations while the generation requires many memory bound decode operations.
> 2. If we are verifying a lot of queries, a probabilistic approach is possible where we only check 10% of the generations. Provided we have sufficient disincentive (e.g. slashing mechanism). The provider is game theoretically incentivized to not risk cheating.
>
> **On providers selectively altering the generation**
>
> If we check each generation, we should be able to catch the provider on the queries that they cheated on. If the user never passes the keyword, we will not be able to detect that the provider has this rule in place, however, in this case, the provider never tampered with the generation and the outputs are correct.
>
> **Weight alterations**
>
> For significant changes, we would be able to detect this as the last hidden activations will be different. Outside the scope of this work, we plan to explore the method's sensitivity to more subtle changes such as the same model after some gradient updates.
>
> **Minor point on Table 1**
> Thanks for the suggestion! We will consider showing proportions instead of absolute counts.

---

### Official Review · Reviewer_WCWr · 2025-03-16

**Overall Recommendation:** 1

**Summary:**

This paper proposes TopLoc, a locality-sensitive hashing-based method for verifying that an output generated by an LLM actually comes from the LLM that the LLM serving provider claims to be using. The author claims that traditional methods for verifying LLM output (e.g., cryptographic approaches or testing model intermediate outputs by a third party) are either computationally inefficient or memory inefficient. The proposed TopLoc method leverages hashing methods to significantly accelerate the verification of LLM output while also reducing memory requirements. Experimental results are presented to demonstrate the effectiveness of the proposed TopLoc method.

**Claims And Evidence:**

The effectiveness claim of the TopLoc method in the submission is supported by the experimental justification. However, the speed and memory usage of the TopLoc method do not seem to be well justified in the experimental evaluations.

**Essential References Not Discussed:**

N/A

**Experimental Designs Or Analyses:**

The experimental design to measure the verification performance of the proposed TopLoc method seems to make sense. Again, it is not clear if the proposed method is really as fast and as memory efficient as claimed. Furthermore, the proposed TopLoc method is not appropriately and sufficiently compared to other baselines for LLM output verification (especially state-of-the-art baseline methods).

**Methods And Evaluation Criteria:**

The evaluation criteria to measure the verification performance of the proposed TopLoc method seem to make sense. However, it is not clear if the proposed method is really as fast and as memory efficient as claimed.

**Other Comments Or Suggestions:**

Please see "Other Strengths And Weaknesses" for more details.

**Other Strengths And Weaknesses:**

Strengths:
- The paper is well-written and well-motivated in general.
- Verifying the output of LLMs and checking if it matches the model provider's claim seems to be an interesting research field.
- The proposed method is easy to follow.

Weaknesses:
- I wonder if there is any trivial solution to solve the motivating problem in this paper, e.g., are users sensitive enough to tell if the quality of the LLM's output changes? If so, wouldn't the users simply tell that the model provider used a different model from what has been claimed in the service? If not, does it really matter which model the model service provider actually uses to serve their customers?
- As discussed above, it is not clear how fast the proposed method is. It is also not clear what the actual memory savings obtained by the proposed method are during LLM output verification.
- The proposed method is not appropriately compared to any prior LLM output verification methods or state-of-the-art baseline methods.

**Questions For Authors:**

- What is the theoretical computation and memory complexity of the proposed method?
- Noting that people tend to chase FP8 precision for LLM pretraining and inference, can TopLoc also work for FP8?

**Relation To Broader Scientific Literature:**

This paper is related to the broader area of verifiable program output, as well as the trustworthiness of large language models.

**Theoretical Claims:**

There is no theoretical claim in this paper.

---

> ### Author Rebuttal · Authors · 2025-04-01
>
> Thank you for taking the time to review our paper. We are glad you found the problem interesting and the proposed method easy to follow.
>
> **Comparisons to previous methods and baselines**
>
> We fully agree that having more comparisons with baselines would be helpful. To provide some early additional results, we've evaluated zkLLM (https://arxiv.org/pdf/2404.16109), SVIP (https://arxiv.org/pdf/2410.22307) and using raw activations. The summary of our experiments are available below. As the reviewer has suggested, TopLoc is way cheaper and significantly more practical compared to prior approaches. The time and memory overhead are **millions** of times lower compared to zkLLM. Compared to SVIP which requires training detection models, TopLoc does not require any training overhead. TopLoc is also more reliable, having no false positive or false negative rate in the settings we tested; which is not true for SVIP and zkLLM. Toploc is also 98,000x less VRAM overhead compared to SVIP.
>
> | |zkLLM|SVIP|Raw Activations|TopLoc|
> |---|---|---|---|---|
> |**Detection model training time**|-|4h21m|-|-|
> |**Commitment size per token**|11MB|20KB|10KB|8B|
> |**Commitment overhead per token**|986s|1.7ms|-|0.26ms|
> |**Validation time**|803s|5.6ms|81ms|81ms|
> |**Provider GPU memory overhead per token**|23.1GB|980MB|10KB|10KB|
> |**FPR**|0%|3%|0%|0%|
> |**FNR (Deterministic)**|0%|4.41%|0%|0%|
> |**FNR (Non-deterministic)**|100%||0%|0%|
>
> **Memory savings**
>
> The 1000x storage efficiency claim in our paper is from comparing TopLoc to storing all the activations directly.
> For example, take the smallest model we tested: Llama-3.1-8B-Instruct, which has a hidden size of 4096. If we stored the final hidden activation for every generated token in bf16, we’d need:
>
> ```
> 4096 elements * 2 bytes = 8192 bytes per token
> ```
>
> With TopLoc, we instead store the top 128 activation values every 32 tokens using a polynomial congruence with 128 coefficients. Each coefficient takes 2 bytes, so the total is:
>
> ```
> 128 * 2 bytes = 256 bytes for 32 tokens → 8 bytes per token
> ```
>
> This is a 1000x reduction.
>
> **On TopLoc’s effectiveness in detecting models inference with FP8**
>
> TopLoc is uniquely effective at detecting precision differences between bf16 and fp32 because of the mantissa check. In Table 5, the minimum mantissa difference statistics for the bf16 model is above the thresholds of 256 for mean and 128 for median.
> This setup is particularly difficult for methods which use a downstream detection model such as SVIP. However, detecting changes in fp8 and lower-bit quantizations (e.g. 4-bit) is more complex (as noted in Section 6.1) and outside the scope of this work, which is already a significant improvement over prior methods.
>
> **The necessity of having algorithms to detect model changes**
>
> > I wonder if there is any trivial solution to solve the motivating problem in this paper, e.g., are users sensitive enough to tell if the quality of the LLM's output changes? If so, wouldn't the users simply tell that the model provider used a different model from what has been claimed in the service? If not, does it really matter which model the model service provider actually uses to serve their customers?
>
> This is a valid question, but detection remains necessary for several reasons:
>
> - Agentic workflows: In many cases, the output may not be directly consumed by a human user but passed into another model or system component. Verifiability must be automated and independent of human judgment to ensure trustworthiness in such pipelines.
>
> - Subtle degradation: Users may not be sensitive to model regressions. A user may accept a suboptimal but satisfactory output without realizing a stronger model would have produced a better response. A user might also wrongly attribute poor performance to the task being too difficult for the model’s capabilities, rather than a degraded model.
>
> - Undetectable biases: Shifts in model behavior (e.g. due to altered prompts or use of finetuned models) can inject subtle biases that are difficult to detect through inspection alone.
>
> **Theoretical Complexity Analysis**
>
> Thanks for bringing this up. We believe it will be a common question for readers, so we will include it in our paper.
>
> As mentioned in **memory savings**, we store 8 bytes per token, which grows linearly with the number of tokens (O(n)). For computation, interpolating a polynomial using Newton’s method has a complexity O(k²), where k is the number of top-k values we use in the proof.
>
> Finally, finding the injective modulus can be done in O(k) time:
> ```python
> def find_injective_modulus(x: list[int]) -> int:
>     for i in range(65536, 2**15, -1):
>         if len(set([j % i for j in x])) == len(x):
>             return i
>     raise ValueError("No injective modulus found!")
> ```
> Although the theoretical worst case constant can be quite large, on average, the function returns in a few iterations. This is because the probability of reaching an iteration decreases exponentially.

---

### Official Review · Reviewer_pCbJ · 2025-03-25

**Overall Recommendation:** 3

**Summary:**

This paper presents TOPLOC, a method to achieve verifiable LLM inference. It uses locality-sensitive hashing for intermediate activations to detect potential unauthorized modifications during the computaion. It uses a polynomial encoding scheme of the memory overhead of proof generation by 1000x.

## update after rebuttal
I have updated my score during the rebuttal. My initial question is the lack of justification of the 1000x speedup (which lets me to doubt several main claims in the paper). In the rebuttal, the author has provided me more detailed calculation, and I am convinced to increase my score to 3.

**Claims And Evidence:**

Yes or No. There are two main claims in the paper:
(1) The method can generate accurate proof, this is justified by the results in Table 1, Figure 3, Table 2.
(2) The method is efficient, it claims to be 1000x more storage efficient, but the reviewer does not observe extensive explanation on how the reduction is calculated (please help kindly correct the reviewer if I understand incorrectly).

**Essential References Not Discussed:**

The references are clear.

**Experimental Designs Or Analyses:**

The experiment design is majorly sound (it analyzes the error with the mentioned dataset and models). However, the experiments are all based on distinguishing fp8 and bf16, which the reviewer is not entirely convinced this is sufficient for the overall claim. For instance, how does the method performs when the model provider uses another smaller model, use 4-bit versus 8-bit (e.g. Several large models, e.g. R1 is in fp8 by default, can the method distinguishes it if the model provider is using 4-bit?)

**Methods And Evaluation Criteria:**

The paper uses UltraChat and several leading SOTA LLMs (Llama-3.1-8B-Instruct, INTELLECT-1-instruct, Gemma-2-9b-it), which the reviewer believes is a good combination for evaluation.

**Other Comments Or Suggestions:**

Please see above comments.

**Other Strengths And Weaknesses:**

Please see the above comments.

The paper is good because it addresses an important problem (it is very likely the current model provider will change the computation to save cost).

**Questions For Authors:**

Please see above comments.

**Relation To Broader Scientific Literature:**

One of the closest papers is zkLLM, where the method seems to make substantial improvement over (if the author can kindly point me to the 1000x calculation).

**Theoretical Claims:**

The reviewer checks the correctness of proof generation and validation. (algorithm 1 and 2).

---

> ### Author Rebuttal · Authors · 2025-04-01
>
> Thank you for your review and for recognizing the importance of the problem our paper addresses.
>
> **Comparison to zkLLM**
>
> To provide some context on the speed and memory claim, we provide some early additional results. Here we evaluated zkLLM (https://arxiv.org/pdf/2404.16109), SVIP (https://arxiv.org/pdf/2410.22307) and using raw activations. The summary of our early experiments are available below. As shown, TopLoc is way cheaper and significantly more practical compared to prior approaches. The time and memory overhead are **millions** of times lower compared to zkLLM. Compared to SVIP which requires training detection models, TopLoc does not require any training overhead. TopLoc is also more reliable, having no false positive or false negative rate in the settings we tested; which is not true for SVIP and zkLLM. Toploc is also 98,000x less VRAM overhead compared to SVIP.
>
> | |zkLLM|SVIP|Raw Activations|TopLoc|
> |---|---|---|---|---|
> |**Detection model training time**|-|4h21m|-|-|
> |**Commitment size per token**|11MB|20KB|10KB|8B|
> |**Commitment overhead per token**|986s|1.7ms|-|0.26ms|
> |**Validation time**|803s|5.6ms|81ms|81ms|
> |**Provider GPU memory overhead per token**|23.1GB|980MB|10KB|10KB|
> |**FPR**|0%|3%|0%|0%|
> |**FNR (Deterministic)**|0%|4.41%|0%|0%|
> |**FNR (Non-deterministic)**|100%||0%|0%|
>
> **1000x more storage efficient**
>
> As mentioned in the table above on comparing against zkLLM, TopLoc is actually millions of times more storage efficient.
> The 1000x storage efficiency claim in our paper is from comparing TopLoc to storing all the activations directly.
> For example, take the smallest model we tested: Llama-3.1-8B-Instruct, which has a hidden size of 4096. If we stored the final hidden activation for every generated token in bf16, we’d need:
>
> ```
> 4096 elements * 2 bytes = 8192 bytes per token
> ```
>
> With TopLoc, we instead store the top 128 activation values every 32 tokens using a polynomial congruence with 128 coefficients. Each coefficient takes 2 bytes, so the total is:
>
> ```
> 128 * 2 bytes = 256 bytes for 32 tokens → 8 bytes per token
> ```
>
> This is a 1000x reduction.
>
> **Detecting changes in inference precision**
>
> TopLoc is uniquely effective at detecting precision differences between bf16 and fp32 because of the mantissa check. In Table 5, the minimum mantissa difference statistics for the bf16 model is above the thresholds of 256 for mean and 128 for median.
> This setup is particularly difficult for methods which use a downstream detection model such as SVIP. However, detecting changes in fp8 and lower-bit quantizations (e.g. 4-bit) is more complex (as noted in Section 6.1) and outside the scope of this work, which is already a significant improvement over prior methods.

---

> > ### Comment · Reviewer_pCbJ · 2025-04-02
> >
> > Thank you for getting back. This addresses my question on 1000x. I will increase my score to 3 to support the paper. Please add the clarification in the final manuscript!

---

> > > ### Author Response · Authors · 2025-04-07
> > >
> > > Thanks for the follow-up and for updating your score. We'll make sure to include the clarification in the final manuscript!

---

### Decision · Program_Chairs · 2025-05-01

**Decision:**

Accept (poster)

**Comment:**

Common concerns included a lack of baselines and proper accounting for memory savings and computational cost analysis. These concerns have largely been addressed in the rebuttal, and are easily integrated into the camera ready. There was one remaining negative reviewer, who did not engage in the discussion. Other reviewers and myself have validated that the reviewer's concerns are addressed. While some questioned the more general problem of the efficiency of verification, this is not specific to this work which improves efficiency over the other reported baselines.

I recommend that this paper be accepted, with the full expectation that the authors will integrate these additional comparisons and clarify the costs in their revision, especially given their importance in this rebuttal period.